

# Regulation of LncRNAs and microRNAs in neuronal development and disease

Cheng Xuan[1], Enyu Yang[1], Shuo Zhao[1], Juan Xu[1], Peihang Li[1], Yaping Zhang[2], Zhenggang Jiang[3] and Xianfeng Ding[1]

[1] College of Life Sciences and Medicine, Zhejiang Sci-Tech University, Hangzhou, Zhejiang Province, China
[2] Department of Oncology, Zhejiang Xiaoshan Hospital, Hangzhou, Zhejiang Province, China
[3] Department of Science Research and Information Management, Zhejiang Provincial Centers for Disease Control and Prevention, Hangzhou, Zhejiang Province, China

## ABSTRACT

Non-coding RNAs (ncRNAs) are RNAs that do not encode proteins but play important roles in regulating cellular processes. Multiple studies over the past decade have demonstrated the role of microRNAs (miRNAs) in cancer, in which some miRNAs can act as biomarkers or provide therapy target. Accumulating evidence also points to the importance of long non-coding RNAs (lncRNAs) in regulating miRNA-mRNA networks. An increasing number of ncRNAs have been shown to be involved in the regulation of cellular processes, and dysregulation of ncRNAs often heralds disease. As the population ages, the incidence of neurodegenerative diseases is increasing, placing enormous pressure on global health systems. Given the excellent performance of ncRNAs in early cancer screening and treatment, here we attempted to aggregate and analyze the regulatory functions of ncRNAs in neuronal development and disease. In this review, we summarize current knowledge on ncRNA taxonomy, biogenesis, and function, and discuss current research progress on ncRNAs in relation to neuronal development, differentiation, and neurodegenerative diseases.

## INTRODUCTION

In 2019, about 50 million people had dementia due to neurodegenerative diseases, and this number is predicted to increase to 1.25 million by 2060 (*World Alzheimer Report, 2019*). Non-coding RNAs (ncRNAs) are involved in various biological processes, including cell proliferation, differentiation, apoptosis, metabolism, stem cell self-renewal, survival and cell integrity maintenance, synaptic formation, and DNA damage responses (*Hombach & Kretz, 2016*). Interestingly, ncRNAs are particularly abundant in the central nervous system, and alterations in their expression pattern have been linked to neuronal differentiation and function. They may lead to brain aging and neurodegenerative diseases (*Mehta, Dempsey & Vemuganti, 2020*). Given the lack of effective treatments for neurodegenerative diseases and the burden on global health systems, early detection, and treatment are required.

Only 1.5% of the human genome encodes protein, and the remaining genes are called "dark matter", which are widely transcribed to generate a massive amount of ncRNAs (*Hombach & Kretz, 2016*). With more and more types of ncRNA being discovered and

Corresponding authors
Zhenggang Jiang, jzgbmu@qq.com
Xianfeng Ding, xfding@Zstu.edu.cn

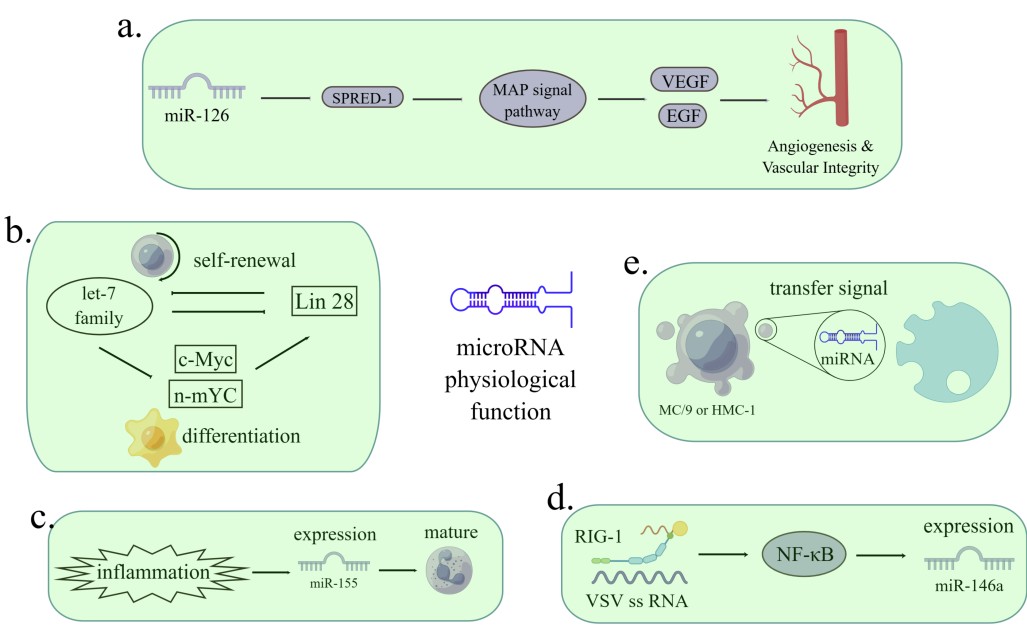

**Figure 1** **The functions of miRNAs in regulating pathological processes.** (A) MiR-126 inhibits the expression of Spred-1, a negative regulator of Ras/MAP kinase signal transduction, thereby enhancing VEGF and FGF and promoting angiogenesis. The loss of miR-126 function reduces the response of MAP kinase to VEGF and FGF, while the function of miR-126 enhances the angiogenesis signal. Spred-1: intracellular inhibitor of angiogenesis signal; VEGF: vascular endothelial growth factor; FGF: fibroblast growth factor. (B) Let-7 family and Lin28 play antagonistic roles in stable self-renewal and differentiation of cells. Let-7 promotes embryonic stem cell differentiation by inhibiting Lin28 and c-Myc. When let-7 is inhibited, cells continue to self-renewal. (C) Under the stimulation of inflammation, the expression of miR-155 is enhanced, thus promoting immature granulocyte numbers in vivo. (D) When infected with a virus, miRNA can either viral target functions and defend against RNA and DNA viruses or can be used by viruses to control cells. After VSV infection, the expression of miR-146a increased in a RIG-1 dependent manner. RIG-1 protein interacts with VSV RNA through its helicase domain, resulting in transcription of pri-miR-1446a in nuclear through NF-$\kappa$ B, leading to an increase in the number of miR-146a. The reciprocity of miRNAs and viruses can be used as a therapeutic target. VSV: vesicular stomatitis virus. (E) MC/9 and HMC-1 can deliver exosomes containing miRNA to other cells, thus transmitting signals. (Figure made with Figdraw).

annotated, many ncRNAs have been proved to control the expression of protein-coding genes and are related to cell cycle, proliferation, differentiation, immune response and apoptosis (*Friedman et al., 2009*). The research on miRNA was relatively earlier and clearer among the various ncRNAs. MicroRNA is widely involved in the physiological and pathological processes of cells. MicroRNA is related to angiogenesis, cell differentiation, inflammatory reaction, virus infection and signal transmission (*Valadi et al., 2007; Wang et al., 2008a; O'Connell et al., 2009; Melton, Judson & Blelloch, 2010; Bruscella et al., 2017*). Figure 1 MiRNAs regulate physiological and pathological processes such as cancer, gastrointestinal diseases, cardiac diseases, diabetes, and liver diseases (*Chen et al., 2019b; Huang, Zhang & Chen, 2022*). The targeting oncogenes of tumor suppressor let-7 include MYC, KRAS and HMGA2, but it act as a tumor promoter when limiting immune cells in tumor microenvironment (*Balzeau et al., 2017; Pobezinsky & Wells, 2018*).

With the discovery of lncRNA, more and more researchers focus on exploiting the functions of lncRNAs and the relationship between miRNAs and lncRNAs (*Xu et al., 2020*). The functions of lncRNA can be divided into five aspects: (i) Location in genomic imprinting; (ii) chromatin modification; (iii) regulation of cell cycle and apoptosis; (iv) regulation of transcription and mRNA decay; (v) regulation of protein translation. Genomic imprinting is an epigenetic phenomenon in which genes are expressed monoallelically based on parent origin (*Bridges, Daulagala & Kourtidis, 2021*). It has been found that multiple lncRNAs are at the imprinted genomic loci, and the loss of imprinting causes abnormal gene expression, resulting in disease (*Zhu et al., 2013*; *Cheong et al., 2015*). HOTAIR, Kcnq1ot1 and Air can recruit chromatin remodeling complexes to silence genes or regulate epigenetics (*Saxena & Carninci, 2011*). The accumulation of Gas5 in growth-arrested cells inhibits glucocorticoid response genes and makes cells sensitive to apoptosis (*Mourtada-Maarabouni et al., 2009*). LncRNA trans-activates STAU1-mediated mRNA decay or targets the sense mRNA transcripts like siRNA (*Gong & Maquat, 2011*). AS-UCHL1 (ubiquitin carboxy-terminal hydrolase L1) significantly increased the synthesis of UCHL1 protein (*Ogawa, Sun & Lee, 2008*). LncRNAs were also involved in diseases such as cancer, nervous system disorders, and other diseases (*Chen et al., 2017*; *Chen et al., 2019a*).

Because both lncRNA and miRNA are closely related to diseases, more and more research is devoted to developing their potential as biomarkers of diseases. However, experiments often require a lot of time and money, which can be solved through computer models. Although computational models have become an essential method for screening the most promising miRNA-disease pairs, their accuracy and universality still need to be improved (*Chen et al., 2018*; *Chen et al., 2019b*). Therefore, computer models can form a reciprocal relationship with experiments. Namely, on the one hand, computer models can guide the most valuable research directions, and on the other hand, experimental results can help optimize computer models (*Chen et al., 2019b*). The LncRNA disease association (LDA) model is similar to the miRNA disease association (MDA) model. Some LDA models are based on classical models, and some implement random forests and feature selection to reduce the interference of noise and redundant information between these data resources (*Yao et al., 2020*; *Cui et al., 2020*; *Wang et al., 2021*). In addition, computer models can also be used to identify new small molecules targeting miRNAs. At present, there are three methods to predict miRNA-associated small molecules: (i) miRNA structure-based models; (ii) models based on gene expression profiles; (iii) known models based on the association of small miRNAs. More effective prediction models will significantly benefit the screening of compound libraries and the discovery of new miRNA-based small-molecule drug candidates (*Chen et al., 2018*).

However, the research on ncRNA function and developing computer models related to ncRNA diseases are focused on cancer. Evidence shows that ncRNAs are closely associated with neurons' development, differentiation, and dysfunction. The human central nervous system has roughly equal numbers of neurons and glial cells, and almost all 86 billion neurons are located in the brain (*Silbereis et al., 2016*). The connectome, one of the most critical components of neural networks and circuits, consists of various neuronal cells and their specific synaptic connections (*Van den Heuvel, Bullmore & Sporns, 2016*). The

human brain weighs 2.5% of the body but still consumes 18% of its oxygen at rest. Humans evolved to accommodate high levels of neuronal activity, including changes in diet and energy allocation, due to the high metabolic cost of the connectome (*Khaitovich et al., 2008*). Some miRNAs contribute to the development of neurons and maintain the survival of mature neurons (*Yoo et al., 2009*). During neuronal differentiation, miR-124 reduces the level of PTBP1, to increase the expression of correctly spliced PTBP2 (*Makeyev et al., 2007*). When cells differentiate into neurons, miR-124 eliminate the biological effects of REST by inhibiting SCP1 (*Shi & Jin, 2009*). LncRNA is involved in many nervous system processes, including neuronal identity establishment and maintenance, stress response deployment, plasticity, and brain development (*Qureshi & Mehler, 2013*). For example, lncRNA Sox2OT is dynamically regulated in the CNS (*Amaral et al., 2009*). REST suppresses the expression of the nervous system-specific transcriptional gene human accelerated region 1 (HAR1), and HAR1 expression changes may be linked to Huntington's disease phenotype (*Johnson et al., 2010a*). The external environment and internal genetic risk factors can lead to neuronal damage and further neuronal degeneration, and when this damage accumulates beyond an individual's "balanced load," neurodegenerative diseases result (*Armstrong, 2020*).

Given the abundant functions of lncRNA and miRNA related to neuronal development and disease, few similar reviews on this topic have simultaneously discussed the relationship between lncRNA and miRNA and neuronal development, differentiation, and disease. We summarized the significance of miRNA and lncRNA in neurodevelopment and disease and their potential role in the future. We want to draw more attention to the potential roles of lncRNA and miRNA in neurons. We believe lncRNA and miRNA can be used as biomarkers and therapeutic targets for some neurological diseases. At the same time, the network basis of lncRNA-miRNA-mRNA can further expand the relevant research direction. However, because there are few research results on ncRNA as a therapeutic target for neurodegenerative diseases or we have yet to be able to retrieve relevant literature, the content of this part needs to be improved, which is the limitation of this review.

## SURVEY METHODOLOGY

After identifying the topic, we searched NCBI for multiple review articles with the keyword "neurons," most of which came from authoritative journals in neurology. In the process of comprehensively reading these review articles, we focused on documenting and listing the outline framework of the articles. We searched the experimental articles according to the content of the framework. When reading experimental articles, we categorize them according to the topics and methods of the study. In the case of controversy in the process of collecting and organizing, we selected articles with a higher impact factor or listed controversial cases. After that, we collected and sorted out the relevant contents and completed the literature review of this article. The parts with similar content are arranged in tables, and pictures supplement the problematic parts to describe them in words.

# NON-CODING RNA AND NEURONAL DIFFERENTIATION

Usually, neuron cells can choose to differentiate or proliferate, which means that when a neuronal cell differentiates, its ability to proliferate is inhibited (*Xie, Sen & Li, 2010*). The Notch signaling pathway involves differentiation and neuronal differentiation (*Bian et al., 2021*). Hes1 is a classic target of the Notch signaling pathway, and high expression of Hes1 inhibits Ascl1 and maintains neural stem cell quiescence. Inactivation of Hes1 and related genes results in the premature termination of neurogenesis, and the healthy activity of neural stem cells depends on Hes1 oscillations (*Sueda et al., 2019*). LncND inhibits neuronal differentiation by sponging miR-143-3p, suppressing Notch protein expression (*Rani et al., 2016*). NBAT-1 inhibits neuroblastoma cell proliferation and promotes neuronal differentiation (*Pandey et al., 2014*). Without SIRT6, H19 inhibits neurogenesis through the p53/Notch1 pathway (*Zhang et al., 2018a*) (Fig. 2).

In addition to controlling neuronal differentiation through the Notch signaling pathway, lncRNAs and microRNAs can also affect other mRNAs, thereby controlling neuronal differentiation. PTBP1 (polypyrimidine tract-binding protein 1) interacts with Pkny, and the knockdown of Pnky or PTBP1 enhances neurogenesis (*Ramos et al., 2015*). LncKdm2b cis-activates Kdm2b and promotes cortical neuronal differentiation (*Li et al., 2020*). LncRNA-1604 orchestrates neural differentiation by competing with the core transcription factors ZEB1 and ZEB2 for miR-200c (*Walgrave et al., 2021*). In the Dicer1-deletion mouse model, miRNAs are verified to be involved in the survival of differentiated neurons, the maintenance of differentiated neuronal cells and brain homeostasis, and the generation of neurons during embryonic cortex generation (*Lau & Hudson, 2010*). MiR-124 promotes neuronal differentiation by targeting DACT1 or Neat1 through the Wnt/β-catenin signaling pathway (*Jiao et al., 2018*; *Cui et al., 2019*).

# NON-CODING RNAS REGULATE NEURONAL FUNCTION

## The effect of ncRNAs on synapse plasticity

Synaptic scale-related protein dysfunction is associated with neurological diseases, and abnormal ncRNA expression is usually associated with protein dysfunction (*Fernandes & Carvalho, 2016*). MiRNA biogenesis is affected by cellular homeostasis, and miRNAs have been reported to be associated with synaptic disorders (*Zheng et al., 2021*). MicroRNAs associated with synaptic plasticity or synaptic disorders are listed in Table 1. LncRNA can control cell differentiation by recruiting transcription mechanisms. KCNA2-AS blocks the recovery of neuronal plasticity after peripheral nerve injury (*Briggs et al., 2015*). ADEPTR deletion can inhibit activity-dependent synaptic transmission changes and dendritic spine structural plasticity (*Grinman et al., 2021*).

## Neuronal degeneration and ncRNAs

Neuronal death caused by progressive neuronal structure or function loss is considered neurodegeneration. Aging, oxidative damage and inflammation are all important factors causing neurodegeneration. Neuronal degeneration caused by environmental toxicants accelerates with aging (*Goedert, Eisenberg & Crowther, 2017*). miR-29 is

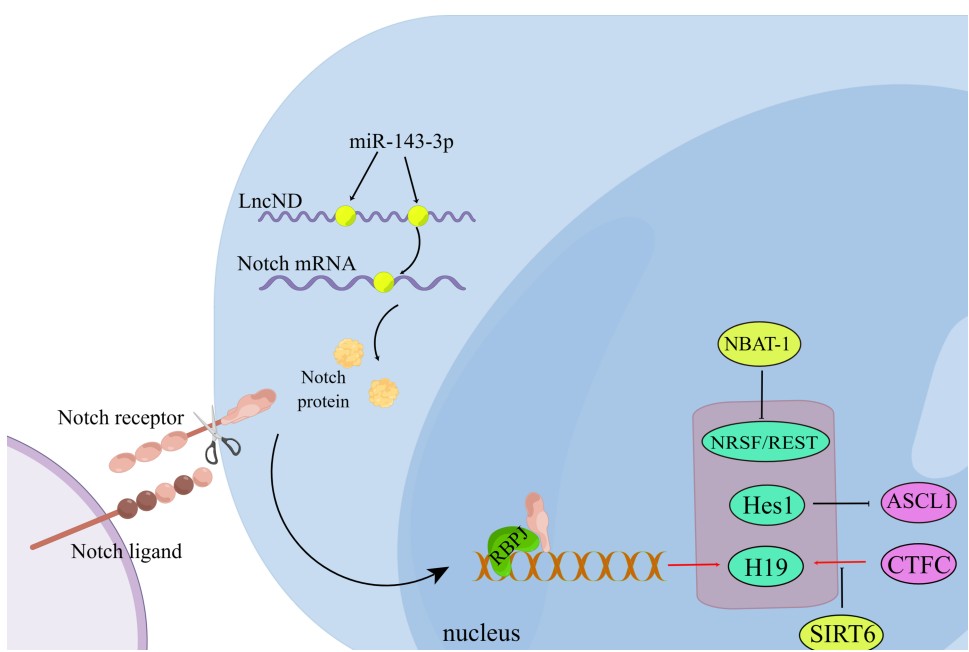

**Figure 2** **Non-coding RNA regulates neuronal differentiation through the Notch signaling pathway.** During the interaction between the Notch ligand and the Notch receptor, the intracellular domain of Notch is cleaved and then transferred to the nucleus, where it interacts with the transcription complex containing RBPJ, resulting in the expression of various Notch target genes. LncND chelates miR-143-3p and releases the expression of *NOTCH* mRNA, thus increasing the production of NOTCH protein required for the maintenance of neural progenitor cells. (Figure made with Figdraw).

adaptively upregulated with aging, and downregulation of the miR-29 family promotes neurodegenerative diseases (*Ripa et al., 2017*; *Jauhari, Singh & Yadav, 2018*). Because the brain has high oxidative metabolic activity but low antioxidant capacity, the brain is highly vulnerable to oxidative stress damage (*Salim, 2017*). H19 targets miR-139 to protect H9c2 cells from hypoxia-induced damage (*Gong et al., 2017*). Inflammation is a significant contributor to cerebral infarction dysfunction. IL-1 stimulates TNFα and IL-1β production by releasing arachidonic acid, which leads to inflammatory aggregation and brain damage (*Zhang et al., 2018b*). MiR-223 deficiency significantly improved clinical symptoms of CNS inflammation, demyelination, and EAE and increased resting microglia numbers and brain microglial autophagy (*Li et al., 2019*).

## Neuronal function and ncRNAs

Much work has demonstrated the relationship between ncRNAs and neuronal function. Dicer deletion reduced the number of mature miRNAs, enhancing learning and memory (*Konopka et al., 2010*). MiR-132 and miR-134 combination might increase the expression of proteins such as Brain-Derived Neurotrophic Factor (BDNF) and Cyclic AMP response element-binding protein (CREB), thereby increasing the formation and maturation of dendritic spines (*Im & Kenny, 2012*). BS-DRL1 interacts with HMGB1 in neurons and regulates responses to DNA damage and genome stability (*Maldonado-Lasuncion et al.,*

**Table 1   MicroRNAs associated with synaptic plasticity or synaptic disorders.**

| MicroRNA | Target mRNA | Functions | Refs |
|---|---|---|---|
| miR-34a | Unknown | Increased miR-34a gene expression may lead to dysfunction of synaptic plasticity, energy metabolism, and resting-state network activity. | *Sarkar et al. (2016)* |
| miR-92a | GluA1(Gria1) | Regulate expression of synaptic GluA1-containing AMPA receptors during homeostatic scaling. | *Letellier et al. (2014)* |
| miR-124 | GluA2 | Express homeostatic synaptic plasticity. | *Hou et al. (2015)* |
| miR-129-5p | Atp2b4 and Dcx | Downregulate Rbfox1 expression and inhibit Atp2b4 and Dcx. | *Rajman et al. (2017)* |
| miR-132 | MMP-9 | Regulate structural plasticity of dendritic spines through MMP-9. | *Jasińska et al. (2016)* |
| miR-135a-5p | Rock2/ Add1 | Loss of miR-135a-5p results in elevated levels of Rock2 and phosphorylation of Ser726 on Add1, resulting in synaptic dysfunction and memory impairment. | *Zheng et al. (2021)* |
| miR-186-5p | GluA2 | Increased synaptic expression of GluA2-lacking AMPA receptors, and block synaptic scaling. | *Silva et al. (2019)* |
| miR-455-3p | Unknown | High levels of miR-455-3p enhances mitochondrial biogenesis, mitochondrial function, and synaptic activity. | *Kumar et al. (2021)* |
| miR-484 | Unknown | Predicted targets of miR-484 were enriched in brain proteins involved in the regulation of synaptic transmission and synaptic plasticity. | *Wingo et al. (2020)* |
| miR-485 | SV2 | Negatively regulate dendritic spine density, PSD-95 clustering, and surface expression of GluR2. | *Cohen et al. (2011)* |

**Notes.**
MMP-9, matrix metalloproteinase-9; Add1, adducin 1; PSD-95, postsynaptic density95

*2019*). Furthermore, lncRNAs were found to serve a functional role in gender differences in depression susceptibility. LINC00473 reduced the amplitude and frequency of sEPSCs only in mPFC pyramidal neurons in female mice and the mPFC and several other forebrain regions in depressed females (*Gururajan, 2020*; *Issler et al., 2020*).

# NON-CODING RNAS IN NEURODEGENERATION

Major neurodegenerative diseases include Alzheimer's disease (AD), Parkinson's disease (PD), Huntington's disease (HD), frontotemporal lobar dementia (FTLD), and amyotrophic lateral sclerosis (ALS). Discrete populations of neurons are lost or damaged in nearly every neurodegenerative disease (*Malhi et al., 2014*). Here, we discussed the relationship between ncRNAs and several common neurodegenerative diseases; other neurodegenerative diseases are listed in Table 2.

## Non-coding RNAs and Alzheimer's Disease

Alzheimer's disease (AD) is the prevalent CNS degenerative disease, which can lead to mood disturbances, cognitive decline and even death (*Kumar, Singh & Ekavali, 2015*). The most striking pathological feature of AD is the "accumulation" of amyloid beta (Aβ) peptides and intracellular neurofibrillary tangles (NFTs) (*Iqbal et al., 2005*). Research shows that ncRNA is associated with increased risk of AD.BACE1 (β-site amyloid precursor protein cleaving enzyme 1) catalyzes APP cleavage to generateβ-amyloid peptides. BACE1-AS and

Peer

**Table 2  Illustrative list of ncRNAs that are disrupted in neuronal disorders.**

| Type | Disease | Involved ncRNAs | Refs |
|---|---|---|---|
| lncRNA | AD | BACE1-AS, GDNFOS, 17A, NAT-Rad18, BC200, Sox2OT, NDM29, 51A, 1810014B01Rik | *Mus, Hof & Tiedge (2007)*, *Parenti et al. (2007)*, *Airavaara et al. (2011)*, *Arisi et al. (2011)*, *Massone et al. (2011)*, *Massone et al. (2012)*, *Ciarlo et al. (2013)* and *Liu et al. (2014)* |
| | ALS/FTLD | C9ORF72, NEAT1-2 | *Nishimoto et al. (2013)* and *Zu et al. (2013)* |
| | AS | UBE3A-AS | *Johnstone et al. (2006)* |
| | Autism | ST7OT (anti-sense to ST7) | *Vincent et al. (2002)* |
| | BWS | LIT1 (anti-sense KvLQT1), Peg8, H19 | *Horike et al. (2000)*, *Okutsu et al. (2000)* and *Sparago et al. (2004)* |
| | GABA neuropathies | Evf-2 (anti-sense Dlx6) | *Feng et al. (2006)* |
| | Fragile X syndrome | BC1 | *Zalfa et al. (2005)* |
| | HD | HDD-AS, HAR1, TUG1, NEAT1, DGCR5, MEG3 | *Johnson et al. (2009)*, *Johnson et al. (2010b)*, *Chung et al. (2011)* and *Johnson (2012)* |
| | Long-term memory disorders | Anti-NOS (anti-sense nNOS) | *Korneev et al. (2005)* |
| | Neuronal hyperexcitability | EVF2 | *Bond et al. (2009)* |
| | PD | Uchl1-AS, PINK1-AS (naPINK1), Sox2OT, BC200, 1810014B01Rik | *Scheele et al. (2007)*, *Carrieri et al. (2015)* and *Luo et al. (2015)* |
| | PWS | UBE3A-AS, IPW, ZNF127-AS (anti-sense ZNF127) | *Wevrick & Francke (1997)*, *Jong et al. (1999)* and *Chamberlain & Brannan (2001)* |
| | Schizophrenia | GOMAFU, DISC2 (anti-sense DISC1), PSZA11q14 (anti-sense DLG2) | *Polesskaya et al. (2003)*, *Millar et al. (2004)* and *Barry et al. (2014)* |
| | Spinocerebellar ataxia 8 | SCA8 (ATXN8OS) | *Daughters et al. (2009)* |
| miRNA | AD | miR-29, miR-146, let-7, miR-9, miR-124, miR-138, miR-181, miR-125, miR-485, miR-107, miR-200, miR-34 | *Scheele et al. (2007)*, *Lu et al. (2008)*, *Nowak & Michlewski (2013)* and *Yang et al. (2017)* |
| | ASD | miR-30 | *Nowak & Michlewski (2013)* |
| | Down's syndrome | let-7, miR-125, mir-155, miR-802 | *Kuhn et al. (2010)* and *Nowak & Michlewski (2013)* |
| | Fragile X syndrome | miR-124, miR-132, miR-125 | *Nowak & Michlewski (2013)* |
| | HD | miR-9, miR-124, miR-132 | *Nowak & Michlewski (2013)* |
| | PD | miR-7, miR-184, let-7, miR-133, miR-34 | *Gehrke et al. (2010)* and *Nowak & Michlewski (2013)* |
| | Rett's syndrome | miR-146a, miR-146b, miR-29, miR-382, miR-132, | *Urdinguio et al. (2010)*, *Wu et al. (2010)* and *Nowak & Michlewski (2013)* |
| | Spinal motor neuron disease | miR-9 | *Haramati et al. (2010)* |
| | Schizophrenia | miR-134, miR-181, miR-219, miR-198 | *Nowak & Michlewski (2013)* |
| | Spinocerebellar ataxia type 1 | miR-19, miR-101, miR-100 | *Haramati et al. (2010)* |
| circRNA | AD | ciRS-7 | *Lukiw (2013)* |
| T-UCR | Idiopathic neurodevelopmental disease | T-UCRs uc.195, uc.392, uc.46 and uc.222 | *Lukiw (2013)* |
| snoRNA | PWS | snoRNA cluster at 15q11–q13 imprinted locus | *Kishore & Stamm (2006)*, *Horsthemke & Wagstaff (2008)* and *Sahoo et al. (2008)* |

**Notes.**

There is not necessarily a clear delineation between classes of non-coding RNA (ncRNA).

AD, Alzheimer's disease; ALS/FTD, Amyotrophic lateral sclerosis and Frontotemporal dementia; AS, Angelman syndrome; BWS, Beckwith–Wiedemann Syndrome; HD, Huntington's disease; PD, Parkinson's disease; PWS, Prader–Willi Syndrome; ASD, Autism spectrum disorders.

miR-485-5p may share the same binding site with the sixth exon of the BACE1 mRNA transcript, preventing miRNA-induced repression of BACE1 mRNA (*Roberts, Morris & Wood, 2014*). MiR-9, miR-125b, and miR-128 are expressed in the Alzheimer's disease brain over the average adult abundance (*Lukiw, 2007*), whereas miR-107 is reduced in early AD and may regulate BACE1 to promote disease progression (*Wang et al., 2008b*). The MiR-29 family is significantly reduced in AD patients, accompanied by abnormally high levels of BACE1 protein (*Shioya et al., 2010*). The neuronal sortilin-related receptor gene (SORL1) interacts with APP in the endosome or trans-Golgi network, affecting trafficking and proteolytic processing, thereby increasing the risk of AD (*Schipper et al., 2007*). 51A reduces the synthesis of SORL1 variant A by driving the SPRL1 splicing shift, thereby impairing APP processing and increasing Aβ production (*Qiu-Lan et al., 2009*).

## Non-coding RNAs and Parkinson's Disease

Parkinson's disease (PD) primarily affects the motor system of the CNS. With the deterioration of PD, there will also be autonomic dysfunction and other non-motor symptoms (*Reich & Savitt, 2019*). Hereditary PD family cases are mainly caused by mutations in the genes SNCA, PARKIN, UCHL1, PINK1, DJ-1, and LRKK2 (*Xiromerisiou et al., 2010*). AS-Uchl1 induces Uchl1 expression by promoting Nurr1 translation. Uchl1 overexpression may be beneficial for treating neurodegenerative diseases (*Carrieri et al., 2015*). The knockdown of NEAT1 significantly inhibits autophagy in PD, thereby attenuating dopaminergic neuron damage (*Yan et al., 2018*). Alpha-synuclein (SNCA) is detrimental to dopamine neurons, both miR-7 and miR-153 effects on SNCA expression additively (*Mouradian, 2012*). Downregulation of miR-34b/c occurred in several brain regions in PD patients, which underlies early mitochondrial dysfunction in PD (*Wang et al., 2008c*). Screening for abnormal expression of ncRNAs in PD patients and models is beneficial to the finding of novel biomarkers or therapeutic targets, but further studies on the pathogenesis of PD are still needed.

## Non-coding RNAs and Huntington's disease

Huntington's disease (HD) is a fatal dominant neurodegenerative disorder induced by repeat expansions of cytosine-adenine-guanine trinucleotides in the Huntington gene. Its symptoms include chorea, mental problems and dementia (*Zuccato et al., 2003*). Overexpression of NEAT1 contributes neuroprotection against neuronal damage in HD through a cell survival pathway under stress conditions (*Choudhry et al., 2015*). Human accelerated region 1 (HAR1) is specifically expressed in the nervous system, and the level of HAR1 in the striatum of HD patients is significantly lower than that of untreated patients (*Sunwoo et al., 2017*). miR-9, miR-9*, miR-29b, and miR-124 are down-regulated in HD, while miR-330 is up-regulated in HD (*Packer et al., 2008*). miR-29a is up-regulated in HD but down-regulated in mouse cortex (*Johnson et al., 2008*). This difference may be due to the different methods used to analyze miRNA expression and continuing problems with RNA quality and integrity in the postmortem human brain (*Johnson & Buckley, 2009*).

## Non-coding RNAs and Other Neurodegeneration Disease

In a mouse model of Amyotrophic Lateral Sclerosis (ALS), an antisense oligonucleotide (ASO) inhibitor of miR-129-5p significantly increases survival and improves the neuromuscular phenotype of treated mice (*Lu et al., 2021*). MiR-155 may be a candidate for co-silencing of miR-129 as its function is mainly involved in CNS inflammation by regulating microglia (*Koval et al., 2013*). A co-role of miR-183/96/182 has been demonstrated in the pathogenesis of ALS/FTD-related aging and cognitive function (*Jawaid et al., 2019*). MiR-146a-5p may play a vital role in regulating neurogenesis in the pathological process of depression. The DG is a critical region for neurogenesis in the adult brain, and microglia-derived exosomes transport miR-146a-5p to the DG region to regulate neuronal function.

## EARLY DIAGNOSIS AND TREATMENT BASED ON MIRNA

Research shows that early diagnosis can reduce the risk of Alzheimer's disease by one-third (*Norton et al., 2014*). However, the existing diagnostic methods are invasive or costly (*Dolgin, 2018*). In addition, due to individual differences, the accuracy of these methods is low (*McKhann et al., 2011*). It is reported that the miRNA profile in brain tissue and blood of patients with neurodegenerative diseases has changed, so miRNA has excellent potential as a biomarker (*Lee et al., 2021*). The cost of new drug discovery is very high, and the period is extended (*Mohs & Greig, 2017*). It is an ideal solution to change the expression of specific miRNAs by inhibitors or endogenous substances or to use miRNA-targeted drug delivery (*Hu et al., 2018*; *Ouyang et al., 2022*). When the expression of miRNAs is changed in cells, it may affect drug sensitivity and regulate drug resistance to standard cancer therapy, thus having a more substantial therapeutic effect (*Adams et al., 2017*).

Despite great potential, the biological application of miRNA still has some insuperable limitations. First, miRNA is prone to degrade *in vivo* due to circulating RNase or cell endocytosis chamber, and its half-life is very short (*Ramachandran & Chen, 2008*). Second, the acceptable delivery methods for human beings are limited. These methods have low penetration efficiency of the blood–brain barrier (BBB) or are unable to target delivery, or the delivery dose is limited (*Lee et al., 2019*; *Ul Islam et al., 2020*). Third, although transfection reagents or nanoparticles can solve the problem of penetrating the blood–brain barrier, the toxicity of small therapeutic oligonucleotides is still a problem (*Ramachandran & Chen, 2008*; *Ruberti, Barbato & Cogoni, 2012*). Last, the relationship between miRNAs and target genes is not always a 1:1 match, and some imitations of miRNAs may lead to t off-target effects (*Loganantharaj & Randall, 2017*).

Despite these limitations, a miRNA is still a powerful tool for the early detection and treatment of neurodegenerative diseases. Thus, we suggest the following points to exploit the potential of miRNA fully. (i) Conduct statistical analysis with the help of bioinformatics to accurately guide the research direction. (ii) Combining with endogenous competitive RNA (ceRNA) mechanism, further improve the linkage map between miRNA and neurodegenerative diseases to seek more accurate targets. (iii) Further modify delivery materials (such as nanoparticles) and oligonucleotides to improve penetration and reduce toxicity.

## CONCLUDING REMARKS

Here, we introduced ncRNAs and the nervous system separately and discussed the roles of ncRNAs in neuronal differentiation, function, and diseases. Although the mechanisms by which ncRNAs affect neuronal function and dysfunction are not fully understood, this review summarizes the current relevant research results. Because the differentiation of the nervous system happens mainly in a specific period, ncRNAs often delay or advance differentiation by acting on specific signaling pathways, thereby affecting the differentiation process. The current findings suggest that neurodegenerative diseases are often the result of disturbances at the protein level. LncRNAs and microRNAs often compete and alter protein stability, thereby changing the ratio of protein concentrations. Due to individual differences and the complexity of the nervous system, there are sometimes two independent studies with opposing results. In the end, our exploration of neuronal remains the tip of the iceberg. It cannot be excluded that with further research on ncRNAs and neurons in the future, more targets will be unlocked for therapeutic purposes.

## ACKNOWLEDGEMENTS

We thank all those who have contributed data and annotation and developed tools and algorithms for ncRNA detection, alignment, and structure prediction. The authors sincerely thank all the participants.

### Funding

This work was supported by the Foundation of Science Technology Department of Zhejiang Province, China social development projects (LGF21C050001). The funders had no role in study design, data collection and analysis, decision to publish, or preparation of the manuscript.

### Grant Disclosures

The following grant information was disclosed by the authors:
Foundation of Science Technology Department of Zhejiang Province, China social development projects: LGF21C050001.

### Competing Interests

The authors declare there are no competing interests.

### Author Contributions

- Cheng Xuan performed the experiments, prepared figures and/or tables, and approved the final draft.
- Enyu Yang performed the experiments, prepared figures and/or tables, and approved the final draft.
- Shuo Zhao performed the experiments, prepared figures and/or tables, and approved the final draft.

- Juan Xu analyzed the data, prepared figures and/or tables, and approved the final draft.
- Peihang Li analyzed the data, prepared figures and/or tables, and approved the final draft.
- Yaping Zhang conceived and designed the experiments, authored or reviewed drafts of the article, and approved the final draft.
- Zhenggang Jiang conceived and designed the experiments, authored or reviewed drafts of the article, and approved the final draft.
- Xianfeng Ding performed the experiments, authored or reviewed drafts of the article, and approved the final draft.

### Data Availability

This is a literature review and does not have raw data.

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
