# Peer review of "Regulation of LncRNAs and microRNAs in neuronal development and disease"

_PeerJ, doi:10.7717/peerj.15197_

## Round 0.1 · original submission · Major Revisions

· Academic Editor

Major Revisions

Thank you for submitting your manuscript to PeerJ, which has been through the peer-review process. Reviewer comments are below. When revising your manuscript, please carefully consider all issues mentioned in the reviewers' comments: please outline every change made in response to their comments and provide suitable rebuttals for any comments not addressed. Please note that your revised submission may need to be re-reviewed.

Reviewer 1 ·

Basic reporting

1. There are some errors about English grammar, spelling and sentence structure in the manuscript. It is suggested that your manuscript should be carefully edited by someone with expertise in technical English.
2. Important reviews about disease-miRNA association identification should be emphatically introduced (PMIDs: 36056743, 36151749, 36094095 and 29045685).
3. This manuscript is difficult to follow due to inaccurate use of words and incomplete structuring of sentences. You should pay much attention to manuscript writing improvement.
4. It is necessary to reference or justify some claims in the paper.
5. As there are many similar studies about analysis of lncRNAs and microRNAs in neuronal development and disease, the introduction section needs to be expanded. The authors should acknowledge much more previous work and point out the novelty of significance of this article.
6. Could you discuss the recent trend of developing computational model for identification of human complex disease-related miRNAs/lncRNAs as the future direction of your current research about neuronal disease-related miRNA/lncRNA biomarker identification?
7. Please describe your opinion on the future direction of this topic in the review.
8. Author should clarify the differences between this review and many previous similar reviews about this topic. In addition, the limitation of this review paper should also be discussed.
9. Two important reviews about lncRNA function and its association with human complex disease should be emphatically introduced (PMIDs: 27345524 and 30247501).

Experimental design

See above

Validity of the findings

See above

Additional comments

See above

·

Basic reporting

1. You should revise your English writing carefully and eliminate small errors in the paper to make the paper easier to understand.
2. Please explain how this manuscript advances this field of research and/or contributes something new to the literature.
3. There were some grammatical errors in the article, and the expression of some of the content was not clear enough. The author needed to check the manuscript carefully and made corresponding revision.
4. Literature review is seriously incomplete. Many important computational models for miRNA/lncRNA-disease association prediction published in the top journals should be discussed and cited, such as the paper with PMIDs: 29045685, 29939227, 30142158, and 24002109. It could be added as an important part of your review.
5. More figures should be provided to help understand the context.
6. What is your improvement over previous review in this field?
7. In recent years, more and more ncRNAs have been identified and increasing evidences have shown that they may affect gene expression and disease progression, making them a new class of drug targets. It thus becomes important to understand the relationship between ncRNAs and drug targets. Could you give some discussions about small molecule drug-microRNA interactions as the future direction of this field (PMID: 30325405)?

Experimental design

no comments

Validity of the findings

no comments

Additional comments

no comments

·

Basic reporting

The Abstract s too basic and vague. It should provide more details of the lncRNAs that are involved in neuronal development and disease. Specific lncRNAs that are discussed in the review should be identified here in the Abstract and the major mechanisms of their actions discussed in the article should also be mentioned here in the Abstract. This is important to provide a snapshot of the article to the readers.

The first paragraph of the Introduction should end at ‘…treatment are required’. The part from ‘Non-coding RNAs act as…’ should be moved after 2nd paragraph because second paragraph introduces the readers to non-coding RNAs and that information needs to come first before this part.

Paragraph 3 of the Introduction deals a lot with ‘cancer’. This review understand that the authors are trying to make a case of lncRNAs-miRNAs connection BUT it would be better to provide examples from neurosciences so that the readers focused on neuronal disease can connect.

The last paragraph of Introduction provides some good information but this seems to emerge out of no where and has no connection with the paragraphs preceding as well no connection with the section following it. Please consider editing it to make connections for the ease of flow of information.

Under section 4, authors mention ‘high impact factor’ – what was the cut off criteria for ‘high impact factor journals’. Also, what was the logic behind such decision. There is obvious bias in the methods and also slightly lower impact factor journals do not necessarily represent unreliable research findings. An ideal approach would have been to include all available data and findings.

Figure 1 is too basic and does not provide any novel information about ncRNAs in neuronal development and disease. If the authors wish, they can retain it. However, another new figure needs to be added that summarizes the information presented and discussed in this article.

Experimental design

The review is appropriately designed and structured.

Validity of the findings

The studies discussed in the article are valid.

---

## Round 0.2 · Major Revisions

· Academic Editor

Major Revisions

1. The manuscript needs extensive English editing. There needs to be more Subject-Verb Agreement in many places in the manuscript.

2. The rebuttal letter should clearly mention the lines where the changes have been incorporated rather than writing paragraphs 1,2,3 etc. It needs to be clarified.

3. In many places, the references need to be included.

4. Reviewer 1: comment 7 and Reviewer 2: Comment 2 should be addressed appropriately.

5. The authors are advised to include this section: Therapeutic strategies using miRNAs or lncRNAs to treat neuronal diseases.

---

## Round 0.3 · accepted · Accept

· Academic Editor

Accept

The authors successfully addressed the majority of concerns raised by the reviewers. Therefore I recommend the article for publication.

Reviewer 1 ·

Basic reporting

It could be accepted

Experimental design

.

Validity of the findings

.

·

Basic reporting

The authors have revised the manusicript according to my suggestions. I accept this manuscript to publish in this journal.

Experimental design

.

Validity of the findings

.

Additional comments

.

·

Basic reporting

Acceptable

Experimental design

Acceptable

Validity of the findings

Acceptable